# Predictive Spike Timing Enables Distributed Shortest Path Computation in Spiking Neural Networks

## Abstract

Efficient planning and sequence selection are central to intelligence, yet current approaches remain largely incompatible with biological computation. Classical graph algorithms like Dijkstra's or A* require global state and biologically implausible operations such as backtracing, while reinforcement learning methods rely on slow gradient-based policy updates that appear inconsistent with rapid behavioral adaptation observed in natural systems.

We propose a biologically plausible algorithm for shortest-path computation that operates through local spike-based message-passing with realistic processing delays. The algorithm exploits spike-timing coincidences to identify nodes on optimal paths: Neurons that receive inhibitory-excitatory message pairs earlier than predicted reduce their response delays, creating a temporal compression that propagates backwards from target to source. Through analytical proof and simulations on random spatial networks, we demonstrate that the algorithm converges and discovers all shortest paths using purely timing-based mechanisms. By showing how short-term timing dynamics alone can compute shortest paths, this work provides new insights into how biological networks might solve complex computational problems through purely local computation and relative spike-time prediction. These findings open new directions for understanding distributed computation in biological and artificial systems, with possible implications for computational neuroscience, AI, reinforcement learning, and neuromorphic systems.

## 1 Introduction

Retrieving sequences and finding shortest paths is fundamental to both artificial and biological information processing systems. In computer science, this is typically achieved using algorithms like Dijkstra's (Dijkstra, 1959) or Bellman Ford (Bellman, 1958; Ford, 1956; Shimbel, 1955; Moore, 1959) and their hierarchical variants (Hart et al., 1968; Geisberger et al., 2008; Funke & Storandt, 2015; Madkour et al., 2017; Dibbelt et al., 2014; Bast et al., 2016; Delling et al., 2009; Blum, 2019). Sequence selection appears across domains including robotics (Kavraki et al., 1996; Latombe, 1991; Lozano-Pérez & Wesley, 1979; Jafarnia-Jahromi et al., 2023), decision making (Puterman, 1994; Chen et al., 2021), language production (Zganec Gros & Zganec, 2008), memory retrieval Wang et al. (2020b), database queries (Griffiths Selinger et al., 1979; Graefe, 1993), hierarchical data traversal (Tarjan, 1972; Cormen et al., 2009), distributed system analysis (Casavant & Kuhl, 1988; Kwok & Ahmad, 1999), and type systems (Pottier & Rémy, 1998; Agesen, 1994).

Modern AI systems epitomize this search paradigm. Transformers function as sophisticated search engines (Vaswani et al., 2017), identifying token sequences that maximize likelihood functions. This highlights that effective search requires both data storage and efficient retrieval mechanisms—a dual requirement fundamental to both artificial and biological intelligence[1]. However, reinforcement learning typically relies on iterative policy updates through gradient-based optimization (Williams, 1992; Sutton et al., 1999). Despite progress in sample efficiency Fatkhullin et al. (2023); Xu et al. (2020); Wang et al. (2024); Ishfaq et al. (2025), this requires extensive trial-and-error learning (Ilyas et al., 2020), which seems inconsistent with the rapid behavioral adaptation observed in biological

---

[1]Also proclaimed in Sutton's *The Bitter Lesson*, 2019.

systems, where animals quickly navigate novel environments (Rosenberg et al., 2021; Vale et al., 2017; Zugaro et al., 2003; Widloski & Foster, 2022; Pfeiffer & Foster, 2013).

The brain must solve similar problems for decision making, memory formation, and path planning. Given abundant dynamically changing sequences in cortical networks (Dragoi & Buzsáki, 2006; Dragoi & Tonegawa, 2011; Dragoi, 2020; Buzsáki & Llinás, 2017; Buzsáki & Tingley, 2018; Diba et al., 2014; Bakermans et al., 2025b; Schwartenbeck et al., 2023) and theoretical frameworks for relational information processing (Whittington et al., 2020; Bakermans et al., 2025a; Waniek, 2020), understanding how the brain implements sequence selection for navigation (Pfeiffer & Foster, 2013) or memory retrieval (Oberauer et al., 2018) is appealing. Yet, how the brain solves this efficiently, while allowing quick, contextually modulated re-planning, remains elusive, with the biggest hurdle being the biological *hardware* and its constraints.

Classical algorithms like Dijkstra's – which is considered optimal for data representable as graphs (Haeupler et al., 2024) – propagate from source to target. Then they use *back-tracing* during which they walk backwards along parent nodes that were collected during the forward search. This paradigm of graph traversal with parent tracking extends to many other algorithms, such as the forward-backward algorithm for Hidden Markov Models (Baum et al., 1970). While computationally efficient on classical hardware, implementing these in biological or neuromorphic systems (Furber et al., 2014; Pei et al., 2019; Davies et al., 2018; Richter et al., 2024; Kadway et al., 2023; Akopyan et al., 2015) faces major challenges. Forward propagation is feasible (Ponulak & Hopfield, 2013; Waniek, 2020; Orsher et al., 2024), but back-tracing requires neurons to remember activation sources. This contradicts evidence that neural dynamics are transient (Friston, 1997; Abeles et al., 1995; Amit, 1997; La Camera et al., 2019; Mante et al., 2013; Ponce-Alvarez et al., 2012; Seung, 1996; Wiltschko et al., 2015; Alderson et al., 2020; Buckley & Nowotny, 2012; Durstewitz & Deco, 2007; Rabinovich et al., 2006; Seliger et al., 2003; Brinkman et al., 2022; Mazor & Laurent, 2005; Koch et al., 2024; Tognoli & Kelso, 2014). Moreover, neural signaling, meaning the transduction of information from one neuron to another, is inherently directional and non-reversible (Maass, 1997; Harris & Shepherd, 2015; Douglas & Martin, 2004; Markov & Kennedy, 2013), precluding straightforward backward information propagation required for back-tracing (and, for that matter, error-backprop and automatic differentiation, but see recent progress in Bellec et al. (2020); Lillicrap et al. (2020); Renner et al. (2024); Whittington & Bogacz (2017); Stanojevic et al. (2024)).

We propose a biologically plausible alternative using spike-timing predictions to compute shortest paths without back-tracing. Our contributions include 1) a novel spike-timing protocol for local shortest path inference, 2) analytical convergence proof, 3) simulation results, and 4) discussion of benefits and limitations. This predictive spike-time paradigm could extend to other algorithms, potentially transforming classical approaches into biologically-plausible, distributed variants that advance both neuroscience and adaptive artificial systems.

## 2 RELATED WORK

Extensive research demonstrates that precise spike timing carries information and drives computation across brain areas (Kayser et al., 2010; Levi et al., 2022; Mainen & Sejnowski, 1995; Montemurro et al., 2007; Shmiel et al., 2006; Rolls et al., 2006; Srivastava et al., 2017), with millisecond-precise patterns encoding stimulus features and behavioral states. Building on this, computational models have proposed temporal coding schemes including rank order codes (Rullen & Thorpe, 2001; Thorpe & Gautrais, 1998) and polychronous networks (Izhikevich & Hoppensteadt, 2009; Szatmáry & Izhikevich, 2010), though these did not demonstrate flexible shortest-path computation. Spike-timing dependent plasticity (STDP) further supports timing-based computation (Masquelier et al., 2008; Bush, 2010; Bi & Poo, 1998; Caporale & Dan, 2008; Gilson, 2010; Markram et al., 2012; Pokorny et al., 2020), operating across timescales for sequence learning and circuit self-organization.

Early theoretical work proposed mechanisms from spike train patterns to synfire chains and gradient fields for sequence generation (Amari, 1972; Abbott & Blum, 1996; Abeles, 1991), with experimental evidence supporting spatiotemporal firing patterns in cortical assemblies (Dabagia et al., 2024; Schrader et al., 2008; Bouhadjar et al., 2022; Abeles et al., 1993). However, these lacked principled shortest-path algorithms with fast goal switching.

Several approaches model sequence learning through reinforcement learning (Samsonovich & Ascoli, 2005) or deep learning techniques for grid and place cells (Banino et al., 2018; Cueva & Wei, 2018; Sorscher et al., 2023), but don't solve shortest paths in biologically relevant timescales. Neural oscillations, especially theta rhythms, organize sequential activity (Wang et al., 2020a; Igata et al., 2021; McNamee et al., 2021; Papale et al., 2016; Parra-Barrero et al., 2021), creating temporal scaffolds for sequence representation and replay phenomena for memory consolidation and planning (Widloski & Foster, 2022; Pfeiffer & Foster, 2013). While demonstrating flexible path computation, the algorithmic mechanisms remain unclear.

Eligibility traces in e-prop (Bellec et al., 2020; Traub et al., 2020) enable biologically plausible back-propagation approximations through local synaptic traces for supervised learning. Our approach instead exploits temporal predictions for unsupervised path-finding without explicit gradients, representing a complementary timing-based paradigm.

Most spiking network path planning approaches use biologically implausible modifications such as weight changes Roth et al. (1997); Ponulak & Hopfield (2013); Schuman et al. (2019), connection removal Davies et al. (2018), or spike-time tables for back-tracing Krichmar et al. (2022). A recent study used spike-threshold adaptation for shortest paths (Dietrich et al., 2025), building on spiking hierarchical temporal memory (Bouhadjar et al., 2022) with separate excitatory/inhibitory populations. While successful in small simulations, their approach differs from ours by modeling distinct populations rather than unified message types, precluding formal convergence analysis and direct biological interpretation.

## 3 Model Description and Convergence Analysis

Our algorithm operates on a network of spiking neurons where each neuron connects to a local neighborhood, enabling message-passing between adjacent nodes. Moreover, neurons can broadcast inhibitory messages globally and exist in one of two meta-states: *tagged* or *untagged*. The core mechanism relies on *predictive tagging*: neurons become tagged when they receive inhibitory-excitatory ($I$-$E$) message pairs earlier than anticipated based on network timing predictions. Initially, only the goal neuron is tagged. When the algorithm begins, excitatory ($E$) messages propagate from the starting neuron throughout the network. Tagged neurons process messages faster and broadcast both local excitatory and global inhibitory signals, causing their predecessors to receive $I$-$E$ pairs earlier than expected. This creates a cascading effect where neurons progressively become tagged, propagating the tagging state backward from goal to source until the shortest path emerges. In the following, we describe the neuron model and provide a formal convergence proof.

### 3.1 Timed State Machine Model of Neural Activity and Neural Interaction

Real neurons typically maintain a resting potential and perform non-linear integration of arriving excitatory inputs until reaching a spiking threshold (Koch, 2004). This threshold depends on factors including neuro-modulation and internal dynamics, while inhibitory inputs can suppress activity and prevent spike generation. We simplify this nonlinear behavior using a timed state machine model where each simulated neuron $v$ exhibits five distinct states: *resting*, *processing*, *spiking*, *refractory*, and *inhibited*. Rather than modeling separate excitatory and inhibitory neural populations following Dale's principle, we represent inhibition and excitation as messages of type $I$ or $E$, respectively, between model neurons.

A neuron remains *resting* until receiving a message, transitioning to *inhibited* ($I$ messages) or *processing* ($E$ messages). An *inhibited* neuron will remain inhibited for a duration of $\tau_{\text{inh}}$, from which it recovers to *resting*. Provided it has not received an $I$ message in the meantime, a *processing* neuron will move to *spiking* after $\tau_{\text{proc}}^0$, and emit an $E$ message to all neurons in its local neighborhood, which incurs another short delay $\tau_{\text{spike}}$. After moving to *refractory*, it will then recover to *resting* after $\tau_{\text{ref}}$. Message transmission incurs axonal delays $\Delta t_I$ and $\Delta t_E$ for $I$ and $E$ messages, respectively, upon sending, and a dendritic delay $\Delta t_{\text{dendritic}}$ upon arrival, with $\Delta t_I < \Delta t_E$ reflecting rapid inhibition (Packer & Yuste, 2011) [2].

---

[2]We further note that biological inhibition often targets the soma directly instead of the dendritic tree of a neuron, which we omit in our simplified model.

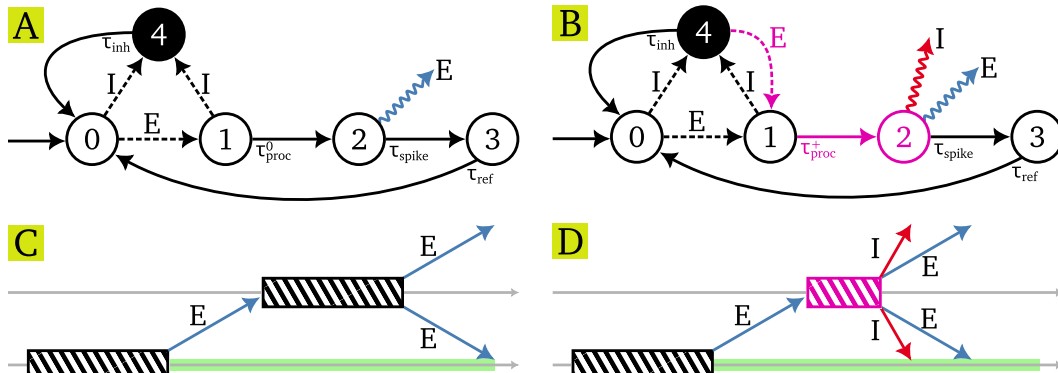

Figure 1: Each model neuron has four states: *resting* (0), *processing* (1), *spiking* (2), *refractory* (3), and *inhibited* (4). A neuron's state machine transits to another state either upon receiving an $E$ or $I$ message (dashed arrows, message type over arrow), or after a temporal delay (solid arrows, delay under arrow). A neuron can emit $E$ and $I$ messages (wavy arrows) when spiking. **(A)** shows the state machine of a *untagged* neuron, which produces only $E$ messages when spiking. **(B)** shows the state machine of a *tagged* neuron, which can move from *inhibited* to *processing*, has a shorter processing delay $\tau_{proc}^+ < \tau_{proc}^0$, and also generates $I$ messages when spiking. **(C)** Sequence diagram of two interacting neurons. After the processing delay (hatched box), the neuron in the bottom row sends an $E$ message (blue arrow) to its neighbors (here only top row), with transmission delay $\Delta t_E$ indicated by the slant of the arrow. It predicts to receive recurrent $E$ messages only after a time window that includes message delivery times and subsequent processing delay (indicated by green bar over the neuron's time axis). **(D)** Sequence diagram where the second neuron (top row) is *tagged*. The tagged neuron has faster processing (magenta hatched box), and emits both $I$ and $E$ messages, with $\Delta t_I < \Delta t_E$ indicated by slant of the arrows. The $I$-$E$ message pair arrives earlier than predicted at the first neuron (bottom row), i.e. within the spike-delay prediction time window.

Tagged neurons exhibit two key differences. First, they transition more rapidly from processing to spiking using $\tau_{proc}^+ < \tau_{proc}^0$, modeling threshold adaptation such as lowering the spiking threshold and local neuro-modulation (Azouz & Gray, 2000; Fontaine et al., 2014; Henze & Buzsáki, 2001). Second, they broadcast global $I$ messages in addition to sending local $E$ messages when spiking, following unspecific inhibition in real networks (Packer & Yuste, 2011). Critically, tagged neurons can transition from *inhibited* to *processing* upon receiving E messages, modeling neuromodulator-gated disinhibition. This enables the backward propagation of tagging states that drives shortest-path discovery.

Figure 1 illustrates the state machines of an *untagged* and *tagged* neuron, and sequence diagrams of neural interactions. While this model substantially simplifies real neuronal dynamics, it captures the essential timing relationships necessary to implement distributed shortest-path computation without requiring global coordination or explicit back-tracing, as neurons self-organize based purely on local timing relationships and predictive coding of spike arrival times.

## 3.2 CONVERGENCE ANALYSIS

To analyze the properties of the distributed, timing-protocol based algorithm, we model the system of interacting neurons as a directed, unweighted graph in which, as introduced above, messages of two distinct types $I$ and $E$ are sent around. Specifically, we have a graph $\mathcal{G} = (\mathcal{V}, \mathcal{E})$, where $\mathcal{V}$ is the set of nodes (or neurons) and $\mathcal{E} \subseteq \mathcal{V} \times \mathcal{V}$ is the set of directed edges between nodes. A path $p$ from node $u$ to node $w$ written $u\text{—}w$, is the sequence of nodes $(u = u_1, u_2, \ldots, u_l = w)$, meaning there exists a directed edge from $u_i$ to $u_{i+1}$, denoted $(u_i, u_{i+1})$ or $u_i \to u_{i+1}$. Two nodes $u, v \in \mathcal{V}$ are connected, if there is a path $u\text{—}w$. The distance $k_{u,v} = d(u, v)$ between two nodes $u$ and $v$ is the minimal number of edges that need to be traversed on any path $u\text{—}w$. The neighborhood $\mathcal{N}(v)$ of $v \in \mathcal{V}$ consists of all nodes $u \neq v \in \mathcal{V}$ which are connected by an edge $v \to u$. The $k$-neighborhood $\mathcal{N}_k(v)$ of $v \in \mathcal{V}$ is the set of all nodes $u \neq v \in \mathcal{V}$ with $d(v, u) \leq k$.

As mentioned above, $v \in \mathcal{V}$ send messages of type $E$ and broadcast messages of type $I$, if $v$ is tagged, after receiving a message of type $E$ themselves with message travel times $\Delta t_I$ and $\Delta t_E$, respectively, and $\Delta t_I < \Delta t_E$. An $I$-message that is sent at time $t$ from node $v$ will thus arrive at all nodes $u \in \mathcal{V}$, whereas an $E$-message will arrive only at nodes $u \in \mathcal{N}(v)$. Without loss of generaltiy, we can omit $\Delta t_{\text{dendritic}}$ in the following analysis.

Recall that at any time $t$ a node $v \in \mathcal{V}$ can be in one of five states and is either tagged or not. If a node is *tagged*, it can emit (forward) messages with short temporal delay $\tau_{\text{proc}}^+$, whereas *untagged* nodes require $\tau_{\text{proc}}^0$ time. Hence, a message of type $E$ that is sent from a tagged node $v$ at time $t$ will arrive at a time $t + \Delta t_E + \tau_{\text{proc}}^+$ at nodes $u \in \mathcal{N}(v)$, while a message of the same type that is sent from an non-tagged node will arrive only at $t + \Delta t_E + \tau_{\text{proc}}^0$ in the neighborhood of the sending node. A message of type $I$ will arrive at all nodes $u \in \mathcal{V}$ at time $t + \Delta t_I + \tau_{\text{proc}}^+$ or $t + \Delta t_I + \tau_{\text{proc}}^0$ after being sent from a tagged or non-tagged node, respectively. A node which is purely *inhibited* and is not tagged itself cannot forward any messages.

A node $v \in \mathcal{V}$ becomes tagged if it produced a message of type $E$ at time $t$ and receives a (recurrent) $I$-$E$ message pair in a shorter-than-expected time window. More precisely, if $v$ receives

1. an inhibitory message $I$ at time
   $t_{I,observed} = t_0 + \tau_{\text{proc}}^+ + \Delta t_I < t_0 + \tau_{\text{proc}}^0 + \Delta t_I = t_{I,\text{expected}}$, and

2. an excitatory message $E$ at time
   $t_{E,observed} = t_0 + \tau_{\text{proc}}^+ + \Delta t_E < t_0 + \tau_{\text{proc}}^0 + \Delta t_E = t_{E,\text{expected}}$, where

$t_0 = t + \Delta t_E$ accounts for the travel time of the $E$-message from $v_i$ to the subsequent node, then $v_i$ itself becomes tagged. In other words, $v$ becomes tagged if it receives an $I$-$E$ message pair in a time window $\Delta t \leq 2\Delta t_E + \tau_{\text{proc}}^+ < 2\Delta t_E + \tau_{\text{proc}}^0$ after sending an $E$ message itself.

In the following, we will show that a system in which a target node $t$ is tagged and the propagation of messages of type $E$ begins at a start node $s$ will eventually converge in finite time to a state where all nodes on a path $p$ from $s$ to $t$ are tagged. We show the convergence and time boundedness using induction, where we show that after $k$ iterations the nodes $k$-closest, i.e. at a distance at most $k$ to the target, are tagged.

**Base Case – Iteration 1** Tagging nodes adjacent to the target $w$, assuming that a path $u$—$w = (u = u_1, u_2, \ldots, u_l = w)$ from starting node $u$ to $w$ exists and that no nodes other than the target $w$ are tagged.

1. (Forward Phase) Let $u, w \in \mathcal{V}$ be the start and the target node, respectively, and let propagation of messages of type $E$ start at $u$. During the first iteration, a message of type $E$ will eventually reach $w$ at time $t$ (after at most $O(N)$ propagations, where $N$ is the number of edges $m$ on the shortest path from $u$ to $w$), and all neighbors of $w$ that are on the shortest path to $w$ will have sent messages of type $E$ and $I$ at time $t_0 = t - \Delta t_E$.

2. (Inhibitory Control, $I$-$E$ pair) At time $t_1 = t + \tau_{\text{proc}}^+$, $w$ sends an $I$-message to all nodes in the network that arrives after $\Delta t_I$ time, and an $E$-message to all its neighbors that arrives after $\Delta t_E$ time.

3. (Tagging Condition) Neighbors $v \in \mathcal{N}(w)$ receive

   (a) an $I$-message at $t_2 = t_1 + \Delta t_I = t + \tau_{\text{proc}}^+ + \Delta t_I$, and

   (b) an $E$-message at $t_3 = t_1 + \Delta t_E = t + \tau_{\text{proc}}^+ + \Delta t_E$.

   Since $t_2 < t_0 + \Delta t_E + \Delta t_I + \tau_{\text{proc}}^0$, $t_3 < t_0 + \Delta t_E + \Delta t_I + \tau_{\text{proc}}^0$, and $t_3 - t \leq \Delta t$, the timing condition for the $I$-$E$ message pair holds. Those $v \in \mathcal{N}(w)$ which sent an $E$-message at time $t$ thus will become tagged, while nodes which did not send an $E$-message will retain their previous state.

**Inductive Hypothesis – Iteration $k$** Assume that after $k$ iterations, all nodes at a distance $k$ from the target $w$ on the shortest path towards the target are tagged.

**Induction Step – Iteration $k+1$**  We need to show that nodes at distance $k+1$ from the target will be tagged after iteration $k+1$.

1. (Forward Phase) In iteration $k+1$, the source node $u$ sends messages of type $E$ that propagate through the network. Nodes at distance $k+1$ from the target $w$ will send an $E$ message at time $t_0 = t - \Delta t_E$.

2. (Inhibitory Control, $I$-$E$ pair) Tagged nodes at a distance $k$ trigger messages of type $I$ and $E$ at time $t_1 = t + \tau_{\text{proc}}^+$, which reach nodes at distance $k+1$ at times $t_2 = t_1 + \Delta t_I$ and $t_3 = t_1 + \Delta t_E$, respectively.

3. (Tagging Condition) Nodes $v \in \mathcal{V}$ at a distance $k+1$ and which are neighbors of a tagged node receive an $I$-message at $t_2$ and an $E$-message at $t_3$. Since $t_2 < t_0 + \Delta t_E + \Delta t_I + \tau_{\text{proc}}^0$, $t_3 < t_0 + \Delta t_E + \Delta t_I + \tau_{\text{proc}}^0$, and $t_3 - t \leq \Delta t$, the timing condition for the $I$-$E$ message pair holds, and all nodes $v$ that sent an $E$-message at $t_0$ become tagged.

Thereby, nodes at a distance $k+1$ from the target along the shortest path are tagged. By induction, the algorithm tags all nodes along the shortest path. $\qquad\square$

## 4  RESULTS

We demonstrate our algorithm's behavior through simulated navigational shortest-path tasks, which are both intuitive to visualize and directly analogous to spatial navigation paradigms used in neuroscience research on place cells and grid cells – specialized neurons in the hippocampal formation that encode spatial information (Moser et al., 2015). Our simulation environment consists of neurons $v \in V$, distributed randomly across a spatial domain while maintaining a minimum packing distance $p_{\min}$ between any pair of neurons to avoid excessive overlap and approximate the distribution of real place cells. Each neuron $v \in V$ is assigned a two-dimensional spatial coordinate $\mathbf{x}(v)$ to determine network connectivity [3]. Specifically, each neuron connects to all neighbors within a spatial annulus, such that $v \in V$ forms connections with all $u \in V$ satisfying $d_{\min}^2 < ||\mathbf{x}(v) - \mathbf{x}(u)||_2^2 < d_{\max}^2$. This connectivity pattern replicates a simplified form of the local transition encoding of grid cells previously proposed by Waniek (2018), or more generally an off-center on-surround receptive field.

All simulations contain 1000 neurons and use $p_{\min} = 0.01, d_{\min} = 0.05, d_{\max} = 0.15$ (in meters). Moreover, $\tau_{\text{proc}}^0 = 10.0, \tau_{\text{proc}}^+ = 5.0, \Delta t_I = 2.0, \Delta t_E = 5.0$ for processing and axonal delays, with additional timing parameters $\tau_{\text{inh}} = 10.0, \tau_{\text{spike}} = 0.1, \tau_{\text{ref}} = 2.0$ and $\Delta t_{\text{dendritic}} = 1.0$ (all times in ms). These parameters were chosen to fall within biologically plausible ranges while providing sufficient network scale and temporal resolution to clearly visualize the algorithm's dynamics and convergence behaviour.

Results for both a square environment and an A-shaped maze are shown in Figure 2. We visualize the algorithm's convergence dynamics as heat maps with overlaid contour lines, where contours encode spike timing relative to algorithm initiation and colors indicate spiking activity and tagging status. This approach enables direct observation of how the *temporal gradient field* evolves as tagged neurons progressively identify the shortest path from source to target in reverse order. To facilitate comparison across iterations, the number of contour levels remains constant throughout each simulation sequence, enabling clear tracking of convergence behavior and emergence of the optimal routing solution.

Initially, neural activity propagates concentrically outward from the starting location, creating circular contours in the temporal gradient field due to uniform processing delays across all neurons. As the algorithm progresses, tagged neurons with accelerated processing times begin to deform this gradient field, biasing the temporal dynamics toward the target location. Global inhibition from tagged neurons suppresses activity along suboptimal paths, effectively pruning exploration in invalid directions. Upon convergence, when the starting neuron becomes tagged, only those neurons that lie precisely on the shortest paths remain active. The algorithm identifies all nodes on shortest paths from start to target, since multiple shortest paths may exist in unweighted graphs, successfully discovering all optimal routes. Additional simulation results for other mazes, different forms

---

[3]Coordinates are only used for setting up the network connectivity and visualization. They are not used in the algorithm.

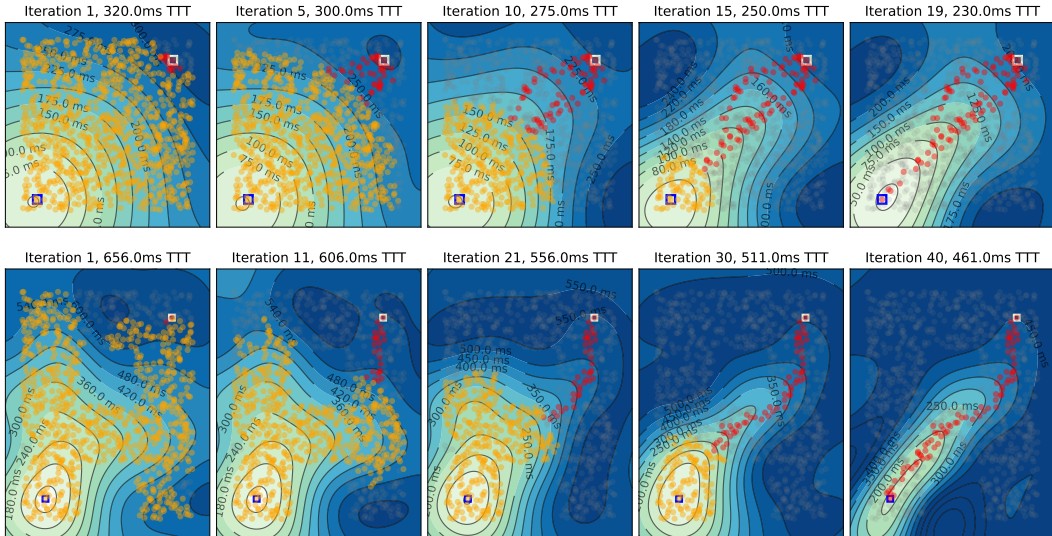

Figure 2: Evolution of the temporal gradient field during iterations of the algorithm. **Top row** shows results for a classical square environment. **Bottom row** shows results for an A-maze, where the spatial domain follows the capital letter A. Starting neuron highlighted by blue box (bottom left corner in each panel), target neuron highlighted by white box (top right). Neurons which spiked during an iteration are colored orange, tagged neurons are red, silent neurons are gray. Contour lines are automatically extracted and indicate time-to-spike for each level. Caption for each panel contains the iteration number, as well as duration during iteration until target neuron spiked. Both rows show the evolution of the temporal gradient field until all nodes on shortest paths are tagged (last column), the decrease in time-to-target (TTT), and the pruning of neural activity along poor directions via global inhibition (silent neurons). The neurons that spike in the final iteration match exactly the shortest-path neurons identified by Dijkstra's algorithm.

of inhibition, and multiple targets can be found in the Appendix. Source code for all simulations is available on github after acceptance.

## 5 DISCUSSION

Our algorithm demonstrates how local spike-timing predictions can solve shortest path problems without global coordination. The key innovation lies in exploiting temporal coincidences of both excitatory and inhibitory inputs. Specifically, neurons become tagged when receiving inhibitory-excitatory message pairs earlier than expected, creating a backward-propagating wave of transient temporal compression from target to source due to threshold adaptation and without learning in form of synaptic weight changes. This transforms graph traversal from a problem requiring explicit backtracing into one of distributed temporal prediction.

While our algorithm operates on general directed graphs, our simulations utilized locally symmetric neighborhoods in unweighted networks to align with grid cell connectivity patterns observed in spatial navigation. Such symmetric connectivity can emerge naturally during exploration, as demonstrated in the Transition Scale-Space (TSS) model (Waniek, 2018; 2020). This framework suggests that grid cells encode transition relationships across diverse topologies – from Euclidean spaces to general Riemannian manifolds – through biologically plausible spatio-temporal learning kernels. Importantly, our core temporal prediction mechanism generalizes to asymmetric and weighted graphs, indicating broad applicability beyond the spatial domain examined here.

While our algorithm guarantees convergence, its runtime scales linearly with path length ($k$ iterations for length-$k$ sequences), limiting efficiency for long routes. The TSS framework demonstrates how multi-scale hierarchical representations can achieve optimal acceleration of retrieval under biologically plausible constraints that match grid cell representations (Waniek, 2020), but does not address spiking implementations. Combining our spike-based temporal prediction mechanism with

TSS's hierarchical scale-space approach represents a natural next step toward achieving both biological realism and computational efficiency. Preliminary work has already demonstrated the feasibility of this integration, showing significantly accelerated convergence using multiple scales while preserving the distributed, local-computation properties of our algorithm.

A critical aspect of our model is the ability of tagged neurons to overcome inhibition and re-enter processing states. We propose this reflects neuromodulator-mediated disinhibition, in which recently active neurons express transient excitability tags that both lower spike thresholds and weaken GABAergic influence through disinhibitory microcircuits (e.g. VIP $\rightarrow$ SOM pathways) and intrinsic channel modulation (Thiele & Bellgrove, 2018). While conceptually related to synaptic tagging and capture mechanisms (Redondo & Morris, 2011) and threshold plasticity (Pham & Hansel, 2023), our tagging operates on much shorter timescales, corresponding to transient dendritic compartment disinhibition that permits rapid re-engagement with ongoing network dynamics.

Our algorithm's backward propagation of tagged states bears striking resemblance to hippocampal replay phenomena, where spike sequences propagate both forward and backward during sharp-wave ripples (Skaggs & McNaughton, 1996). Recent evidence shows replay occurs not only for experienced trajectories but also for novel, never traversed paths (Gupta et al., 2010; Denovellis et al., 2020), and even during route planning as *preplay* (Pfeiffer & Foster, 2013). These findings suggest the brain may indeed use shortest-path dynamics for path optimization.

Our algorithm demonstrates robust convergence in idealized conditions. Several factors could compromise performance in biological settings, such as network noise, imprecise timing, and heterogeneous neural properties, all of which may degrade the temporal prediction accuracy required for proper tagging. The algorithm's reliance on precise temporal windows means that jitter in synaptic delays or variability in processing times could lead to false tagging or missed detection of early $I$-$E$ message pairs. However, this jitter could be exploited in future work to further encode path probabilities and other relative information. Additionally, the global inhibition mechanism may become metabolically costly in very large networks, as tagged neurons must broadcast to all other neurons simultaneously. This broadcast requirement could also create informational bottlenecks, particularly in densely connected networks where multiple tagged neurons generate overlapping inhibitory signals. We note that alternatives for inhibition are certainly possible in our algorithm, for instance requiring more than a single node for sending $I$ messages. However, this would require a more elaborate mechanism for inhibitory control, which we excluded for the sake of keeping the timing protocol and its analysis as simple as possible. Furthermore, the algorithm assumes a static network topology during convergence, making it potentially vulnerable to dynamic environments where connections change or obstacles appear during path computation. These limitations suggest that biological implementations may require additional robustness mechanisms, such as temporal averaging of tagging signals or hierarchical organization to reduce global communication overhead, which need to be explored in future work.

Our model makes several testable predictions for biological and behavioral experiments. During spatial navigation tasks, we predict that neurons closer to the goal should show progressively shorter response latencies in successive trials, reflecting the backward propagation of temporal compression from target to source. This latency gradient should emerge even for novel paths, distinguishing our mechanism from simple experience-dependent plasticity. Additionally, neurons on the optimal path should exhibit characteristic timing behavior that differs from neurons on suboptimal routes. In fact, optogenetic experiments could directly test the role of inhibitory signaling by selectively disrupting inhibitory interneurons, which should impair path optimization without preventing basic path finding, while enhancing inhibitory drive should accelerate convergence. Furthermore, our model predicts that replay sequences during rest periods should exhibit the temporal compression signature of tagged neurons, with backward replay showing faster propagation along previously optimized routes. Single-cell recordings could reveal neurons that can overcome inhibition when tagged, displaying reduced spike threshold and altered temporal dynamics. These predictions provide concrete experimental avenues for testing whether biological path-finding systems employ temporal prediction mechanisms similar to our proposed algorithm.

The prevalence of disinhibitory mechanisms in cortical circuits (Reimann et al., 2024; Sridharan & Knudsen, 2015; Williams & Holtmaat, 2019) suggests our approach could be extended to more elaborate timing protocols. Future work should explore how multiple temporal scales and more complex inhibitory dynamics could enable broader classes of graph algorithms to be implemented

through local spike-timing predictions, potentially revealing fundamental principles underlying the brain's remarkable computational capabilities.

## 6 CONCLUSION

We have demonstrated that biologically plausible shortest path computation can emerge from purely local spike-timing dynamics. Our algorithm exploits predictive spike-time coding, where neurons become tagged upon receiving message pairs earlier than anticipated, creating temporal compression that propagates backward from target to source. This transforms distributed graph search into a problem of local temporal prediction.

The biological plausibility stems from reliance on established neural mechanisms such as spike-timing prediction, threshold adaptation, and competitive inhibition. Unlike traditional algorithms that require global state management, our method operates through local message-passing with realistic delays, making it implementable in both biological circuits and neuromorphic hardware. The formal convergence guarantees demonstrate that this is a principled solution, not merely a heuristic.

Our approach to predictive spike time dynamics represents a paradigm that could extend to other algorithms, potentially transforming classical approaches into biologically plausible, locally distributed variants. This could advance both neuroscience and artificial systems for search, planning, and decision making, highlighting that the brain's computational prowess may rely on elegant temporal dynamics and timing codes that emerge from basic neural properties. These principles could also inspire new AI architectures that combine biological realism with computational efficiency.

### USE OF LARGE LANGUAGE MODELS

LLMs have been used to polish writing, shorten paragraphs for length constraints, and improve sentence flow while preserving all technical and logical content and references.

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

## A  Appendix

## B  Additional Simulation Results

The following contains additional simulation results, in which we explored the algorithm behavior for other environments that are classically used in neuroscience and behavioral studies (Figures 3 and 4, what happens when inhibition is not global but local (Figure 5, when disabling all inhibition which will lead to algorithm failure (Figure 6), and finally the behavior of the algorithm for multiple targets (Figures 7, 8, and 9).

The information presented in the figures follows Figure 2. That is, the starting neuron, meaning the neuron from which activity starts to spread out, is indicated as a blue box and is in all simulations in the bottom left corner for consistency. Each target neuron is indicated by a square white box underlying the neuron itself. The topographical map indicates the temporal landscape of neural activity, and the number of contour levels is kept fixed for each simulation to allow better comparison across iterations. The alpha value of a neuron – and therefore its color intensity – indicates whether a neuron spiked during an iteration or not.

The results show that the algorithm works also with only local inhibition, as long as there is only one target neuron, but shows significantly more remaining activity in the network (Figure 5). Results with multiple simultaneous targets indicate interesting characteristics of the algorithm, in particular in Figure 9, which shows that two of the three selected target neurons establish tagged neurons along a shortest path, but only one remains in the end. Future work will have to explore if additional properties of shortest paths can be encoded in specific relative timing events, and if multiple shortest paths can be successfully super-imposed within one neural population, maybe into different cycles of an oscillation.

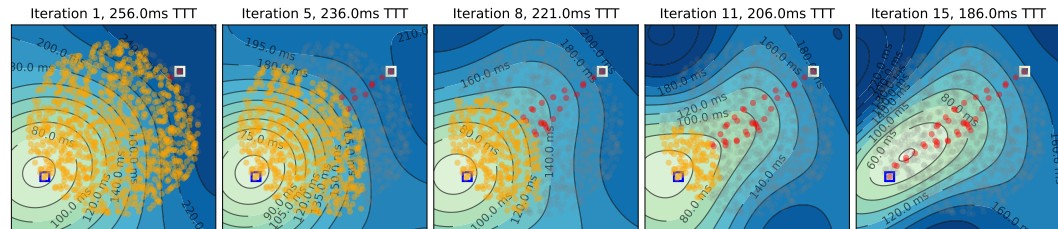

Figure 3: Evolution of the temporal gradient field during iterations of the algorithm in a circular maze. Clearly visible is the concentric propagation of activity during the first iteration (left panel), which over time turns into an elongated ellipse from start towards the target (right panel). This deformation of the contour lines shows the deformation of the temporal gradient field. That is, while the first iteration has neurons that operate alike and therefore propagate activity uniformly in time, the faster *tagged* neurons during the final iteration stretch the temporal gradient from start towards target and block activity on poor paths via inhibition.

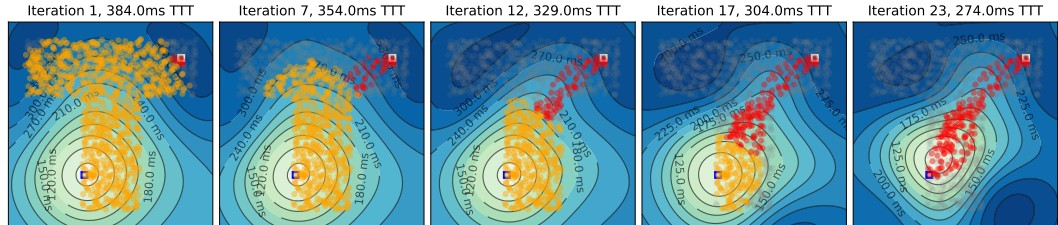

Figure 4: Evolution of the temporal gradient field during iterations of the algorithm in a T-Maze. Due to the high density of neurons in the maze as well as the shape of the maze, there is not a single (or approximately two, as in Figure 3), shortest path but several neurons that are on a broader shortest *band* towards the target. This is expected, given that distance between neurons is considered to be *unweighted* in this implementation of the algorithm.

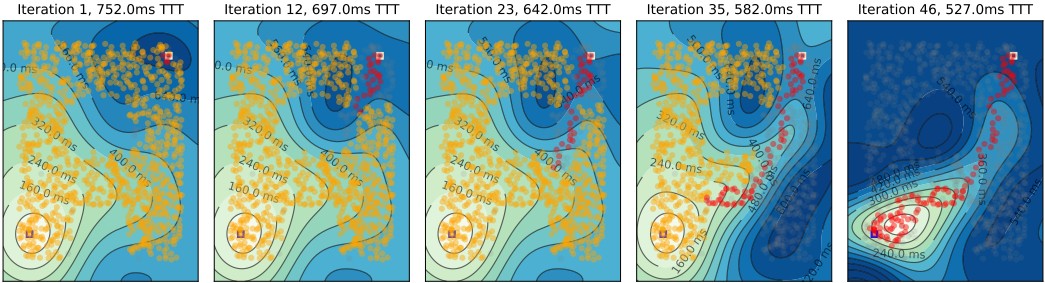

Figure 5: Evolution of the temporal gradient field during iterations of the algorithm in a A-Maze without global inhibition, but with local inhibition. In this simulation, global inhibition was turned off in favor of local inhibition by tagged neurons. Specifically, the local inhibition followed the same connectivity profile as the local excitation, i.e. an annulus around a neuron. The simulation shows that significantly more activity remains active within the network compared to Figure 2, and sub-optimal routes only get extinguished once the starting neuron is tagged. Afterwards, only tagged neurons remain active, and the shortest path appears during the final iteration of the algorithm.

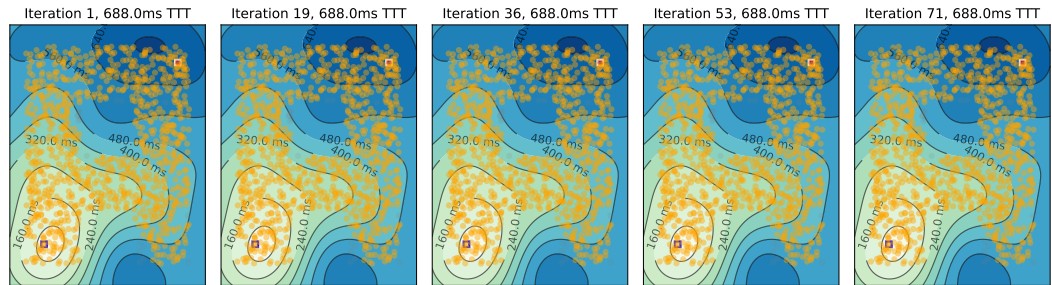

Figure 6: Algorithm failure due to missing inhibition. Without any form of inhibition, the algorithm fails to tag neurons, and the shortest path will not emerge.

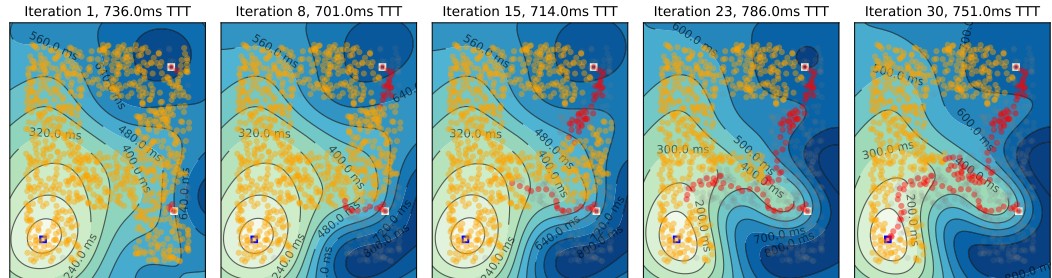

Figure 7: Simulation results for 2 targets and only local inhibition. During this run of the simulation, we disabled global inhibition in favor of local inhibition (see caption of Figure 5 for more details), but added an additional target neuron in the bottom right area. During the final iteration of the algorithm, both trajectories are active. The reason is that neurons on the shortest paths to either target are close enough in the middle section of the paths, so that activity from one shortest path reaches the other. Due to missing global inhibition, which could prevent the secondary shortest path, the activity continues until all paths are found.

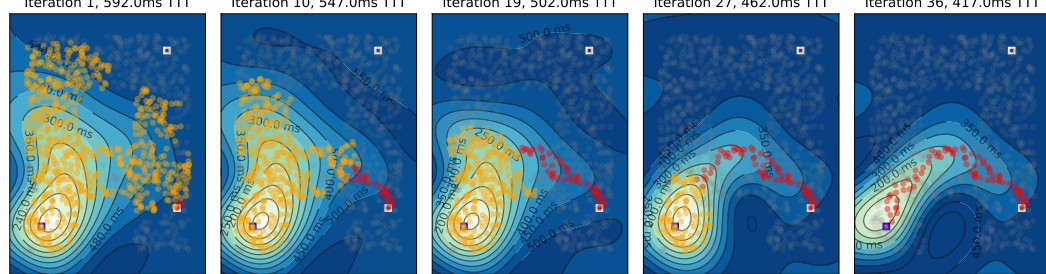

Figure 8: Simulation results for 2 targets and global inhibition. In contrast to Figure 7, we enabled global inhibition. In this case, the global inhibition prevents further propagation of activity towards the top-right target early on. This stops neurons in that direction from getting tagged, and therefore not second shortest-trajectory can emerge in the first place.

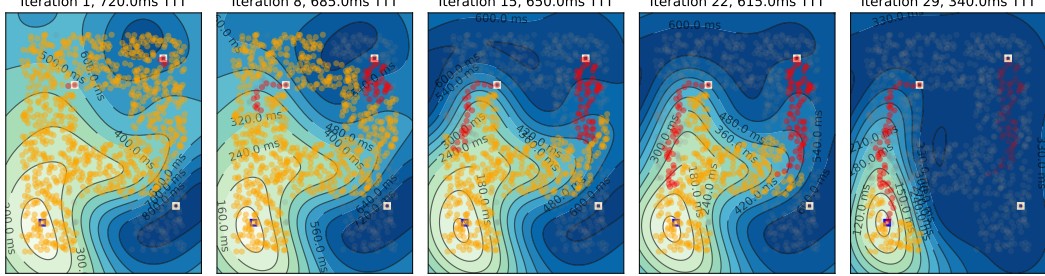

Figure 9: Simulation results for 3 targets, and only local inhibition. During the first iterations, shortest paths towards two of the three targets start to form. After some further iterations, the local inhibition along the trajectory to the target in the top left area extinguishes further excitatory propagation to the secondary remaining shortest path.

