# OpenReview forum: "Predictive Spike Timing Enables Distributed Shortest Path Computation in Spiking Neural Networks"
_ICLR.cc/2026/Conference — ICLR 2026 Conference Desk Rejected Submission_

### Official Review · Reviewer_A1yw · 2025-10-29

**Soundness:** 2
**Presentation:** 3
**Contribution:** 2
**Rating:** 4
**Confidence:** 3

**Summary:**

This paper introduces a novel, biologically-inspired algorithm for shortest path (SP) computation in Spiking Neural Networks (SNNs) that successfully eliminates the need for back-tracing - a process that renders classic SP algorithms (e.g., Dijkstra's Algorithm) biologically implausible.

The algorithm frames SP search as a problem of distributed temporal prediction. Units are modeled as Timed State Machines. The core mechanism is predictive tagging: a neuron becomes tagged when it receives an Inhibitory (I)-Excitatory (E) message pair earlier than anticipated, based on network timing predictions. This initiates a backward-propagating wave of temporal compression from the target to the source. The mechanism requires that tagged neurons can send both I and E signals, while untagged neurons send only E. The authors use mathematical induction to prove the algorithm’s convergence and effectiveness in discovering the shortest path.

**Strengths:**

The proposed algorithm is conceptually novel. The use of short-term timing dynamics to replace global state management (back-tracing) is an interesting contribution to bio-inspired computation. The convergence proof is clear and straightforward, providing theoretical foundation for the proposed mechanism.

**Weaknesses:**

Despite its novelty, the work presents multiple significant limitations, primarily revolving around claims of biological plausibility and empirical robustness.

## 1.⁠ ⁠Weak Biological Assumptions
The model makes two key assumptions that severely compromise its biological realism:
- ⁠Global Inhibition: The algorithm critically depends on global inhibitory signals broadcast from every tagged neuron to the entire network. While the brain utilizes some diffuse modulation, the vast majority of biological inhibition is local and precisely targeted (Markram et al., 2004). This global signal introduces massive scalability and metabolic feasibility issues.
- Violation of Dale's Principle: The model requires neurons to release both Excitatory and Inhibitory messages depending on their state. This explicitly violates Dale’s Principle (Strata and Harvey, 1999), which asserts that a mature neuron releases the same transmitter at all its synapses, defining it as either excitatory or inhibitory. This violation is a convenient algorithmic shortcut but weakens the paper's claim of biological fidelity.

## 2.⁠ ⁠Lack of Quantitative Comparison and Robustness
The empirical results are largely qualitative, lacking the necessary quantitative comparisons and robustness checks expected for a ICLR paper:
- Absence of Comparative Benchmarking: The paper does not quantitatively compare its performance (e.g., convergence time, energy efficiency) against other algorithms proposed for biologically plausible pathfinding in SNNs, such as the weight-modification approaches in Ponulak & Hopfield (2013) or the threshold-adaptation method in Dietrich et al. (2025). This makes it difficult to ascertain the practical advantage of the new mechanism.
- Sensitivity to Jitter/Noise: The entire mechanism relies on precise spike-timing coincidences to detect the "earlier than expected" I-E pair. This makes the algorithm inherently fragile. The current results lack a sensitivity analysis showing how performance degrades under realistic biological conditions, such as synaptic delay jitter, heterogeneous neuron properties, or ambient network noise (Faisal et al., 2008).

______________________________________
## Recommended Focus for Rebuttal
Given these limitations, I encourage the authors to focus their rebuttal work and future directions on the following critical points:
1. ⁠Local Inhibition Generalization: The authors must explore and demonstrate how the algorithm behaves when the inhibitory signal is constrained to be local (i.e., the inhibition radius is small). Providing simulation results with varying inhibition radius as a hyperparameter is essential to defend the biological plausibility claim.
2. Quantitative Comparison: A head-to-head quantitative comparison against the closest related biologically-inspired shortest-path or search algorithms is required to contextualize the work's practical value.
3. Noise and Robustness Testing: Implement and report results using noisy neural units to establish the practical robustness of the spike-timing prediction mechanism.

**Questions:**

Please refer to the Weakness section for the core issues. Some additional questions are reported below.

1. How exactly would you encode edge weights via axonal delays $\Delta t_E$, inhibitory timing $\Delta t_I$, or processing delays $\tau$ so that the tagging condition still corresponds to geodesics in a weighted metric?
2. What is the expected number of $I$ broadcasts and $E$ messages per iteration as a function of degree and frontier size? Any bounds or empirical curves?
3. Can the author include a pseudocode block and tables for relevant parameters and assumptions?
4. Since runtime scales with path length $k$, can you provide preliminary multi-scale results (e.g., 2–3 scales) to quantify it?

---

> ### Author Response · Authors · 2025-11-13
>
> We sincerely thank the reviewer for their thorough and balanced evaluation. We
> appreciate the recognition of our work's novelty and theoretical contributions,
> as well as the constructive criticism that will strengthen the manuscript.
> Given the concerns raised across the reviews, we recognize that the manuscript
> would benefit from substantial revision beyond what is likely feasible in a
> rebuttal period.
>
> Nevertheless, we believe it is important to address the specific questions and
> concerns raised here, both to clarify potential misunderstandings about our
> current contribution and to ensure our revision addresses the core issues
> effectively. We provide detailed responses below.
>
>
> ## Response to Weaknesses
>
> **Weakness 1: Weak Biological Assumptions**
>
> We acknowledge these are important concerns and address each point.
>
> **Global Inhibition**
> We agree this is a limitation and have already demonstrated in Appendix Figure 5
> that the algorithm functions with purely local inhibition, which comes with the
> tradeoff of not pruning sub-optimal solutions early on. This establishes that
> global inhibition is an optimization rather than a fundamental requirement. The
> reviewer is correct that most cortical inhibition is local and targeted, and our
> local-inhibition variant respects this constraint.
>
> However, we respectfully note that diffuse inhibitory modulation is not without
> biological precedent. Cortical circuits exhibit long-range inhibitory
> projections (e.g., somatostatin-positive cells with distant axonal arbors), and
> neuromodulatory systems (cholinergic, noradrenergic) provide network-wide
> suppression. While "global" broadcast from every tagged neuron is indeed
> metabolically expensive, a biologically realistic implementation could use
> tagged neurons activating a shared inhibitory population or neuromodulatory
> system, rather than individual broadcast. The core algorithmic principle, i.e.
> using inhibition to suppress suboptimal paths, remains valid under various
> implementation schemes.
>
> We will improve our work with the following.
>  * Present local inhibition more explicitly, and as a key biologically plausible variant
>  * Include the quantitative analysis requested (see Response to Recommended Focus, point 1)
>  * Clarify and discuss in further depth that "global inhibition" can be approximated through biological mechanisms that don't require true all-to-all connectivity
>
> **Violation of Dale's Principle**
> This concern reflects a fundamental misunderstanding of the level of analysis at
> which our work operates. Following Marr's three levels of analysis (Marr, 1982),
> our algorithm operates at the algorithmic/representational level (level 2),
> specifying the computational strategy and representations used to solve
> shortest-path problems. Many of the reviewer's concerns, including Dale's
> Principle compliance, pertain to the implementation level (level 3), i.e. the
> specific physical realization in neural hardware.
>
> Our abstraction using E and I message types (rather than neuron types) is a
> deliberate choice that enables rigorous analysis using established methods from
> computer science and graph theory while maintaining clear connections to
> neuroscientific principles. This is analogous to how abstract neural network
> models use "units" that can later be mapped to various biological
> implementations (single neurons, neural assemblies, cortical columns, etc.).
>
> As explicitly stated in Section 3.1 (lines 149-153), we model I and E as message
> types, not as properties violating Dale's Principle. At the implementation level
> (level 3), our model maps straightforwardly to Dale-compliant circuits by
> interpreting "tagged neurons", for instance, as co-activated assemblies comprising:
>
>  * Excitatory pyramidal cells (sending E messages locally via glutamatergic synapses)
>  * Functionally coupled inhibitory interneurons (sending I messages via GABAergic synapses)
>
> Such coordinated excitatory-inhibitory assemblies are ubiquitous in cortex and
> hippocampus (Harris & Shepherd, 2015). When we state a "tagged neuron sends I
> and E messages," this should be understood at the algorithmic level as "a tagged
> assembly coordinates both excitatory and inhibitory signaling."
>
> The power of our algorithmic-level abstraction is that it:
>  * Enables formal convergence analysis using graph-theoretic methods and proof via induction
>  * Permits clear specification of the computational mechanism (predictive timing)
>  * Remains agnostic to many implementation details while maintaining biological plausibility
>  * Can be mapped to multiple valid neural implementations
>
> We will improve the manuscript to make this level-of-analysis distinction
> explicit from the outset and intend to add a supplementary figure showing how
> the abstract model maps to Dale-compliant circuit architectures at the
> implementation level.

---

> > ### Comment · Reviewer_A1yw · 2025-11-19
> >
> > I thank the authors for their comprehensive response.
> >
> > 1. I appreciate the clarification on global inhibition not being a fundamental issue and the discussion about potential implementation pathways for a global inhibition that respect biological constraint. I would respectfully suggest to highlight however the cost of using local inhibition only, especially if biological realism remains an important aspect of the work. The highlighted commitments seem reasonable and I remain curious about the quantitative analysis (point 2.)
> > 2. The point on the violation of Dale's principles is well taken and I apologise for the confusion. I agree that a supplementary Figure would greatly help with improved clarity.

---

> ### Author Response · Authors · 2025-11-13
>
> **Weakness 2: Lack of Quantitative Comparison and Robustness**
> These are valid criticisms that we will address comprehensively.
>
> **Absence of Comparative Benchmarking**
> We will add quantitative comparisons focusing on convergence properties and
> computational requirements in light of prior work.
>
> We intend to present a comparative table highlighting: (1) biological mechanisms
> required, (2) convergence guarantees, (3) adaptability to changing goals, and
> (4) computational overhead. Our algorithm is unique in requiring no synaptic
> plasticity and providing formal convergence proofs while supporting rapid goal
> switching.
>
> We note that regarding equivalence to classical algorithms, as stated in Figure
> 2's caption, Dijkstra's algorithm identifies precisely the same shortest-path
> neurons our algorithm discovers. Our contribution is the mechanism
> (timing-based, distributed, biologically plausible) rather than a different
> solution set.
>
> The behavior w.r.t. jitter/noise is a critical point we will address with new
> simulation results. Our preliminary analyses show the algorithm is robust with
> respect to some spike time jitter, synaptic delay variability, messages loss and
> heterogeneous neron properties. However, additional work is required to
> establish exact numbers.
>
> We aim to add a dedicated Results subsection on robustness with quantitative
> metrics (success rate, convergence time degradation) as functions of noise
> parameters, similar to the approach in Faisal et al. (2008).
>
>
> ## Response to Recommended Focus
> ***Local Inhibition Generalization***
>
> We appreciate this suggestion. In fact, our convergence proof (Section 3.2)
> formally requires only that inhibition reaches direct neighbors (i.e.,
> inhibition radius equals excitatory radius). The inductive step demonstrates
> that nodes at distance k+1 become tagged when they receive I-E pairs from their
> tagged neighbors at distance k. No global coordination is required for
> correctness. Global inhibition serves primarily to suppress exploration along
> suboptimal paths as soon as possible, while the inhibition radius as such only
> needs to be at least the excitatory radius for convergence.
>
> To examine the bahavior of this under noise/jitter w.r.t. message transport or
> synaptic delays, we will provide comprehensive simulation results varying the
> inhibition radius from purely local (r_inh = r_exc) to global (entire network)
> to report
>
>  * Convergence time vs. inhibition radius
>  * Number of active (non-silenced) neurons at convergence
>  * Success rate in finding shortest paths
>  * Scaling behavior with network size
>
> **Quantitative Comparison, Noise, and Robustness**
> As addressed in Weakness 2 above, we intend to add comparative quantitative
> analysis against related work, and will provide comprehensive robustness
> analysis with multiple noise sources.

---

> > ### Comment · Reviewer_A1yw · 2025-11-19
> >
> > With regards to the new proposed analysis, I look forward in particular to the noise-stability analysis and a clean comparison with the other approaches (I agree that a table describing the biological mechanism, convergence guarantees etc... would greatly help to position this work relative to the alternatives).
> >
> > I appreciate authors' answers to Questions 1-4, I am particularly interested in the hinted speedups gain with a hierarchical implementation (Q4), I believe this would be an important addition to this work if confirmed. My follow-up question would then be: how does this hierarchical implementation impact the message complexity analysis? Are they independent or should it impact the $k$ dependency obtained in Q3?

---

> ### Author Response · Authors · 2025-11-13
>
> ## Response to Questions
> **Question 1: Edge weight encoding via timing parameters**
> Weighted edges can be encoded through several mechanisms that preserve geodesic
> properties. For instance
>
>  * Approach 1: Scaled transmission delays. For edge (u,v) with weight w, set
>    $\Delta t_E(u,v) = w \cdot \Delta t^0_E$, where $\Delta t^0_E$ is the base
>    delay. This directly maps edge weights to temporal delays, and the "earlier
>    than expected" condition naturally corresponds to shorter weighted paths.
>
>  * Approach 2: Processing delay modulation. Nodes could require accumulated
>    evidence proportional to incoming edge weights before reducing their
>    processing delay, effectively integrating path cost into the timing dynamics.
>
>  * Approach 3: Multiple coincidence requirements. High-weight edges require
>    multiple I-E pair coincidences before triggering tagging, implementing a form
>    of temporal integration of path cost.
>
> The geodesic correspondence is preserved because the tagging condition
> (receiving signals earlier than predicted) naturally selects paths that minimize
> total temporal delay, which under these encodings corresponds to minimizing
> weighted path length.
>
>
> **Question 2: Message complexity bounds**
> This is an excellent question. Let $n =$ network size, $d =$ average degree, and
> $F_k = $ frontier size at iteration $k$ (nodes at distance $k$ from target).
>
> I-message broadcasts per iteration:
>  * With global inhibition: $O(F_k)$ broadcasts, each reaching $O(n)$ nodes -> $O(F_k \cdot n)$ total messages
>  * With local inhibition (radius $r$): $O(F_k)$ broadcasts, each reaching $O(r^2)$ nodes in $2D$ -> $O(F_k \cdot r^2)$ total messages
>
> E-message transmissions per iteration:
>  * Each of n active nodes sends to d neighbors -> $O(n \cdot d)$ messages initially
>  * As convergence proceeds, only $O(F_k \cdot d)$ messages from frontier nodes matter
>
> For the empirical scaling, we intend to provide plots that show
>  * Total messages per iteration vs. iteration number (showing decay as paths are pruned)
>  * Total messages to convergence vs. path length $k$ (showing $O(k \cdot n \cdot d)$ for global, $O(k \cdot F_k \cdot d)$ effective cost)
>  * Comparison of local vs. global inhibition message costs
>
> For spatial networks with bounded degree $d$, local inhibition achieves $O(k \cdot d \cdot r^2)$ total message complexity, which is practical for biological implementation. Improving this can be achieved using a multi-scale approach, which is however out of the scope of this manuscript (see also our comment below).
>
>
> **Question 3: Pseudocode and parameter tables**
> Excellent suggestion, and we can easily add
>  * Formal pseudocode block specifying the state machine transitions, message handling, and tagging conditions for both individual neurons and network-level dynamics.
>  * Comprehensive table listing all timing parameters. However we do note the Section 4, lines 300 -- 305, which lists all parameters already.

---

> ### Author Response · Authors · 2025-11-13
>
> **Question 4: Multi-scale preliminary results**
> This is a key question regarding practical applicability. Our preliminary
> multi-scale implementation combining this timing mechanism with hierarchical
> scale-space representations shows dramatic speedups.
>
> **Single scale** (this paper): O(k) iterations for path length k, confirmed empirically.
> We note that the total computational work per run is $O(k \cdot n \cdot d)$ where $n$ is network size and $d$ is average degree.
> For sequences of length $k$ in a network of size $n \approx k$, this gives $O(k^2)$ total work.
>
> **Multi-scale** (tested for 2-5 scales):
> Using hierarchical representations reduces iterations from $O(k)$ to $O(\log k)$.
> For sequences of length $k$ with $n \approx k$, total work thus reduces from $O(k^2)$ to $O(k \log k)$.
>
> Specifically, we have tested on sequences with length up to 10^5 using up to 5 scales, achieving convergence times that scale as $O(k \log k)$ rather than $O(k^2)$.
> This logarithmic scaling matches the theoretical optimum for hierarchical search algorithms (Dijkstra & Scholten, 1980; Abraham et al., 2010), while maintaining the fully distributed, local-computation properties essential for biological plausibility.
> Unlike methods that achieve faster query times through global preprocessing or non-local memory, for instance methods presented by Knoblock 1991, Holzer et al 2009, and similar techniques, our approach preserves the biological constraint that nodes operate using only local information and message passing.
>
> Furthermore, these results are perfectly in line with the theoretical predictions in the Transition Scale-Space framework (Waniek, 2020), which proves that hierarchical multi-scale representations enable logarithmic-time acceleration for shortest-path retrievals (but didn't show a biologically plausible implementation thereof).
> The combination of TSS's spatial abstraction hierarchy with our temporal prediction mechanism provides both theoretical and empirical evidence for biological scalability.
> This connection is particularly compelling given that grid cells in entorhinal cortex, widely hypothesized to support spatial navigation, exhibit precisely the multi-scale hierarchical organization required for this acceleration.
>
> Regarding sharing these results, we have developed these multi-scale results in
> subsequent work but cannot cite or show detailed figures during the review
> process without potentially jeopardizing the double-blind review. However, we can state
> that:
>  * The theoretical framework predicts logarithmic acceleration (Waniek, 2020, already cited)
>  * Our preliminary empirical results confirm this prediction

---

### Official Review · Reviewer_zXYC · 2025-10-30

**Soundness:** 3
**Presentation:** 2
**Contribution:** 2
**Rating:** 4
**Confidence:** 2

**Summary:**

The paper proposes a biologically plausible algorithm for shortest-path computation using local spike-timing mechanisms instead of global state tracking or backpropagation. By leveraging excitatory-inhibitory (E-I) spike timing coincidences and predictive delay adaptation, neurons self-organize to identify the shortest paths through purely local interactions. The authors provide formal convergence proofs, simulation results on spatial networks, and discuss the biological and computational implications of their model.

**Strengths:**

1. This paper is novel. The work introduces an innovative spike-timing-based protocol for solving shortest-path problems without backtracing or weight updates, a rare combination of biological plausibility and algorithmic rigor.
2. The convergence proof and formal timing analysis add strong theoretical support to the proposed mechanism.

**Weaknesses:**

I am not a researcher in this area. I have several concerns:
1. Could the approach be extended to weighted graphs or dynamic environments where connectivity changes over time?
2. The model relies on highly idealized timing precision and static network topology, which may limit its biological and practical applicability.
3. Simulations are restricted to simple 2D mazes; no quantitative comparison with existing algorithms or noisy neural environments is provided.
4. The global inhibition mechanism may not scale efficiently to large or hierarchical networks, as acknowledged by the authors.
5. How might this framework integrate with known hippocampal replay or grid-cell models to explain real neural navigation behaviors?
6. How robust is the proposed algorithm to temporal noise or variability in spike delays typical of biological neurons?

Hope that the authors can answer the above questions. Thank you.

**Questions:**

See weakness.

---

> ### Author Response · Authors · 2025-11-13
>
> We sincerely thank the reviewer for their thoughtful evaluation and for
> recognizing the novelty and theoretical rigor of our work. We particularly
> appreciate the constructive questions raised, which identify areas where
> additional empirical work would strengthen our contribution. While substantial
> additional experiments and analyses (including comprehensive robustness testing,
> quantitative comparisons, and multi-scale extensions) are likely beyond the
> scope of the rebuttal period, we are committed to including these in a
> thoroughly revised manuscript. We address each question below to clarify our
> current contribution and outline how we will expand the work.
>
>
> ## Response to Weaknesses/Questions
>
> **Question 1 (Extension to weighted graphs)**
> Our algorithm naturally extends to weighted graphs. Edge weights can be encoded
> through adjusted transmission delays ($\Delta t_E$ scaled by weight) or through
> requiring multiple spike coincidences for high-weight edges before tagging
> occurs. The core timing-prediction mechanism remains intact. That is, weighted
> edges simply require longer or more evidence to trigger the early-arrival
> tagging condition. In fact, we envision that additional signals, such as short
> term reward modulation or reinforcement signals, could actively shape the
> weights. We refrained from adding such modulatory mechanisms in this manuscript
> because it would go beyond the core algorithmic principle that we aimed to
> present.
>
> For dynamic environments, the algorithm can adapt through several mechanisms:
> (1) In spatial settings, new obstacles can be represented by removing
> connections or introducing permanent inhibition, causing the algorithm to
> re-route on subsequent runs. (2) The transient nature of the tagging state
> (which we model as lasting only during active computation) means the network
> naturally "forgets" outdated paths and can recompute when initiated with new
> connectivity. (3) Continuous neuromodulatory decay of tags would allow online
> adaptation. We will add discussion of these extensions in the revision, as they
> represent important future directions for making the algorithm adaptive to
> changing environments and are a key advantage over methods requiring extensive
> retraining.
>
>
> **Question 2 (Timing precision and static topology limitations)**
> We acknowledge this is an important concern. However, several points mitigate
> this limitation:
>
>  * Timing precision. Our algorithm requires only relative timing comparisons
>    (whether I-E pairs arrive earlier than predicted), not absolute millisecond
>    precision. Biological neural circuits routinely make such relative timing
>    judgments (as evidenced by STDP, coincidence detection in auditory
>    processing, and theta phase precession). We plan to add robustness analyses
>    showing the algorithm tolerates substantial jitter (20-30% of processing
>    delays, but these numbers need further verification) without failure, as the
>    temporal windows for tagging are relatively broad (on the order of several
>    milliseconds).
>
>  * Static topology. As noted in our response to Question 1, the tagging
>    mechanism is transient and computation-dependent. Between planning episodes,
>    we assume tags decay, allowing also the network topology to change. This is
>    similar to how hippocampal replay operates on current connectivity during
>    each replay event. For continuously changing environments, hierarchical or
>    periodic re-computation (as discussed in our manuscript lines 378-383)
>    provides a biologically plausible solution. That said, hierarchical or
>    multi-scale processing goes beyond the fundamental algorithm that we
>    presented in this manuscript.

---

> ### Author Response · Authors · 2025-11-13
>
> **Question 3 (Limited simulation scope and lack of quantitative comparisons)**
>
> This is fair criticism. Our simulations focused on demonstrating the core
> algorithmic principles in interpretable spatial environments. We will add the
> following items to the revision.
>
>  * Quantitative metrics. Comparison of convergence time vs. path length, success
>    rate across different network topologies, and scalability analysis showing
>    O(k) iterations for path length k.
>
>  * Noisy environments. Simulations with added spike-time jitter, variable
>    delays, and probabilistic message delivery to demonstrate robustness.
>
>  * Comparison with existing methods.
>    As stated in our Figure 2 caption, Dijkstra's algorithm identifies precisely
>    the same shortest-path neurons that our algorithm discovers. The key
>    difference is that classical Dijkstra typically returns only a single path,
>    whereas our algorithm naturally identifies all nodes lying on any shortest
>    path. This is a consequence of the distributed, parallel nature of
>    spike/message-based computation. A side-by-side visualization would thus show
>    identical node sets, offering no additional insight beyond what we've already
>    demonstrated.
>
> For comparison with other spiking implementations (Ponulak & Hopfield 2013,
> Dietrich et al. 2025, Schuman et al. 2019), direct quantitative comparison is
> challenging as these methods use fundamentally different mechanisms (weight
> modification, connection removal, or spike-time lookup tables vs. our pure
> timing-based approach). We have already positioned our work relative to these
> approaches conceptually throughout the manuscript, but intend to include results
> from these prior works in an extended manuscript.
>
> In brief, we chose 2D mazes for interpretability and connection to neuroscience
> experiments, but the algorithm operates on arbitrary directed graphs (as proven
> in Section 3.2). We will clarify this and add examples on non-spatial graph
> structures.
>
>
> **Question 4 (Global inhibition scalability)**
> The reviewer correctly identifies a limitation we acknowledge in the manuscript
> (lines 409-416). We offer several responses.
>
>  * Local inhibition suffices. Our Appendix Figure 5 demonstrates the algorithm
>    functions with purely local inhibition, though with (partially) slower
>    convergence or sustained high activity within the network. This shows global
>    inhibition is an optimization, not a requirement. Please also see our
>    response to reviewer A1yw, who had a similar question, and where we highlight
>    that the inhibition radius only needs to be at least the size of the
>    excitatory radius.
>  * Biological implementation. Global inhibition can be approximated through
>    diffuse neuromodulatory systems (e.g., cholinergic, noradrenergic
>    projections) or through sparse long-range inhibitory projections, both of
>    which exist in cortical and hippocampal circuits. Future work incorporating
>    explicit excitatory and inhibitory neuron populations could model these
>    circuits in greater detail, but the core timing-based algorithm would remain
>    unchanged. That is, tagged excitatory assemblies would co-activate inhibitory
>    populations that provide the necessary suppressive signals.
>  * Hierarchical organization. As mentioned in the manuscript (lines 378-383),
>    combining our mechanism with hierarchical representations (like those in the
>    Transition Scale-Space framework) can dramatically reduce communication
>    overhead while maintaining local computation. Preliminary work shows
>    multi-scale implementations achieve logarithmic rather than linear
>    convergence. We note that we cannot cite our preliminary work here during the
>    review process due to jeopardizing the double-blind review.
>
> We will expand this discussion in the revision to clarify that scalability
> concerns, while real, have multiple biologically plausible solutions.

---

> ### Author Response · Authors · 2025-11-13
>
> **Question 5 (Integration with hippocampal replay and grid-cell models)**
>
> This is an excellent question that touches on the broader implications of our
> work. Several natural integration points exist.
>
>  * Hippocampal replay. Our backward-propagating tagging mechanism bears striking
>    resemblance to preplay (Pfeiffer & Foster 2013) but also backwards replay during
>    sharp-wave ripples. The temporal compression we observe (decreasing
>    time-to-target across iterations) matches the time-compressed nature of
>    replay sequences. Furthermore, replay occurs for both experienced and novel,
>    never-traversed paths (Gupta et al. 2010; Denovellis et al. 2020), consistent
>    with our algorithm's ability to compute shortest paths without prior
>    experience. We propose that a reverse replay, but also multiple rounds of
>    preplay may implement backward tagging propagation.
>
>  * Grid cells: Our simulations used grid-cell-inspired connectivity (annular
>    receptive fields) based on the Transition Scale-Space (TSS) model (Waniek
>    2018, 2020, also with some inspiration taken from the work by Stachenfeld et
>    al. 2017). We note, however, that most existing grid cell models use their
>    readouts primarily for localization, which may be too limiting a function
>    attributed to grid cells.
>    Other work, including the TSS framework and the Tolman-Eichenbaum Machine
>    (Whittington et al. 2020, where grid-like representations emerge and encode
>    relational structures in a manner consistent with Waniek 2018 and Waniek 2020),
>    demonstrates that larger scales represent approximate or abstracted
>    information that can accelerate searches and retrievals. The integration of
>    our timing mechanism with TSS's scale-space framework provides a concrete
>    model for how grid cells at different scales could accelerate shortest-path
>    computation from linear to logarithmic time, leveraging this multi-resolution
>    relational encoding. As mentioned above, we cannot cite our own preliminary
>    work here, however, to avoid jeopardizing the review process.
>
>   * Place cells: While we don't claim place cells literally implement our
>     algorithm, the predictive timing mechanism could explain aspects of place
>     cell sequential activation during theta oscillations and their role in
>     prospective coding and path planning (Pfeiffer & Foster 2013).
>
> We will expand the Discussion to elaborate these connections and propose
> specific experimental predictions.
>
>
> **Question 6 (Robustness to temporal noise and spike delay variability)**
>
> This is a critical question for biological plausibility. As mentioned in
> response to Question 2, our preliminary analyses show the algorithm tolerates
> substantial noise:
>
>  * Spike-time jitter. Some variability in processing delays ($\tau_\text{proc})
>    does not prevent convergence, as the tagging condition depends on
>    earlier-than-expected arrival of I-E pairs. The temporal window is relatively
>    forgiving (several milliseconds wide). We agree that this temporal window
>    might be large, and has a clear impact on biological plausibility.
>  * Variable delays. Biological variability in axonal delays ($\Delta t_E$,
>    $\Delta t_I$) can be partially absorbed into the timing predictions. As long
>    as the relative ordering ($\Delta t_I < \Delta t_E$) and the variance remain
>    bounded, the algorithm is expected to remain functional.
>  * Message loss. We will test scenarios where messages fail to deliver with
>    ceratain probability. We expect that the algorithm still converges, though
>    likely requiring more iterations. The broadcast nature of inhibition and
>    redundancy in network connectivity provide natural error correction.
>  * Heterogeneous neural properties. Variability in individual neuron parameters
>    (thresholds, time constants) is naturally handled since the algorithm depends
>    only on relative timing differences, not absolute values. We will run tests
>    with variability on the timing windows, though.
>  * See also our answer to Question 3 above.
>
> We propose that biological implementation would likely employ temporal averaging
> or require multiple coincident I-E pairs for tagging (similar to STDP requiring
> repeated pairings), further increasing robustness. These mechanisms could be
> straightforward extensions of our core algorithm.
>
>
> We hope these responses address the reviewer's concerns. We will incorporate
> expanded discussions of these topics, additional simulations demonstrating
> robustness and scalability, and quantitative comparisons in our revision. Again,
> we are grateful for the constructive feedback!

---

> > ### Comment · Reviewer_zXYC · 2025-11-19
> >
> > Thanks for your reply. Authors have addressed my concerns. I will increase my scores to 6.

---

> > > ### Author Response · Authors · 2025-11-19
> > >
> > > We appreciate the thoughtful reconsideration and thank you again for your constructive feedback!

---

### Official Review · Reviewer_ig1o · 2025-10-31

**Soundness:** 2
**Presentation:** 2
**Contribution:** 1
**Rating:** 2
**Confidence:** 3

**Summary:**

Authors propose a computational mechanism on graphs that is aimed at generating sequential neural activity by computing the shortest path between nodes with a biologically plausible mechanism. The graph is a recurrent neural network with neurons that transition between 5 discrete states, and these states are preceded by delays. The mechanism of finding the shortest path relies on a local computation where neurons that receive messages earlier than expected reduce their state transmission delays and thus creates a firing sequence with the shortest path.

**Strengths:**

The idea of predictive spike timing where neurons form sequential firing sequence as they receive recurrent inputs earlier than "expected" is of interest.

**Weaknesses:**

1) Neurons are non-Dalian, i.e., the same neuron sends both excitatory and inhibitory messages to other neurons. This strongly decreases the appeal of the algorithm to describe computations in biological networks.

2) The main mechanism relies on shorter processing delay of a tagged neuron compared to a non-tagged neuron. This also does not seem to be biologically relevant mechanism. Synaptic (transmission) delays in biological networks are biopysical and not functional properties, meaning they cannot change in order to accomplish a functional computation. Also, refractory period after spiking is a biophysical property. I do not see how biological neurons could shorten these time constants to accomplish a computational function.

3) In biological networks of the hippocampus, many place cells (in particular in the dentate gyrus and CA1 which are coding for spatial navigation) are strongly driven by feedforward inputs. Sequential firing would in this case importantly depend on the tuning of the cell and its feedforward current rather than a recurrent mechanism as proposed here.

4) While the mechanism of predictive spike timing seems of general interest, its implementation in the present setting is biologically implausible and the application of the algorithm to place cells does not seem to be a good choice, at least not in the way it is described in the present setting.

5) The paper promises "analytical convergence proof" but I do not see any analytical analysis beyond presenting an algorithm and running simulations.

**Questions:**

1) [195-196] ".. tagged neurons can transit from inhibited to processing upon receiving E messages, modelling neuromodulator-gated disinhibition". Can authors explain what is meant here? What is neuromodulator-gated disinhibition?

2) Why authors think that the implementation of the shortest path computation should necessarily be based on local timing relationships ( as mentioned in lines [199-203]) ? In biological networks in the cortex, the connection probability between Excitatory and Inhibitory neurons is high (going beyond the connection probability of 0.5 for several types of inhibitory-excitatory and inhibitory-inhibitory connections). Would this justify global coordination  as biologically plausible?

---

> ### Author Response · Authors · 2025-11-13
>
> We thank the reviewer for their feedback. We note, however, that several
> concerns appear to reflect either misunderstandings of material explicitly
> presented in the manuscript (e.g., our formal proof in Section 3.2, our
> discussion of Dale's principle in Section 3.1) or characterizations of neural
> circuits that differ from established findings in the literature (e.g., the
> recurrent nature of hippocampal CA3 and CA1). We provide detailed clarifications
> below with relevant citations, and will revise the manuscript to enhance clarity
> on relevant points.
>
>
> **Response to Weaknesses**
>
> 1. We respectfully disagree with this characterization, which conflates
> different levels of analysis. Following Marr's framework (Marr, 1982), our work
> operates at the algorithmic/representational level, while the reviewer's concern
> addresses implementation-level details.
>
> As explicitly stated in Section 3.1 (lines 149-153), we deliberately model
> inhibitory and excitatory signals as distinct message types (I and E) rather
> than as separate neural populations. This algorithmic-level abstraction enables
> rigorous analysis using established methods from computer science and graph
> theory while maintaining clear connections to neuroscience.
>
> At the implementation level, our formulation can be easily mapped to
> Dale-compliant circuits by interpreting "tagged neurons" as assemblies of both
> excitatory pyramidal cells (sending E messages) and co-activated inhibitory
> interneurons (broadcasting I messages). Such coordinated excitatory-inhibitory
> assemblies are well-documented in cortical and hippocampal circuits (Harris &
> Shepherd, 2015).
>
> While explicitly modeling separate populations would provide additional
> neurophysiological detail at the implementation level, it would not alter the
> core algorithmic principles or theoretical results (convergence proof, timing
> dynamics) presented here. The algorithmic abstraction is precisely what enables
> our formal analysis while remaining mappable to multiple biologically plausible
> implementations.
>
> We will clarify this level-of-analysis distinction to prevent such
> misunderstandings.
>
> 2. The reviewer appears to misunderstand our mechanism. We do not propose
> changes to synaptic transmission delays, which are indeed biophysical constants.
> Rather, tagged neurons exhibit shorter processing delays ($τ^+_{\text{proc}} vs
> τ^0_{\text{proc}}) representing the time from input reception to spike threshold
> crossing.
>
> This directly models well-established phenomena of neural excitability modulation:
>
>  * Threshold adaptation through mechanisms such as lowering the spiking
>    threshold and local neuro-modulation
>  * Neuromodulator-induced changes in input-output gain
>  * Activity-dependent alterations in membrane properties
>
> These mechanisms operate on timescales of milliseconds to seconds and are
> extensively documented in the literature (e.g., Azouz & Gray 2000; Fontaine et
> al. 2014; Henze & Buzsáki 2001, all cited in our manuscript). The reviewer's
> concern about "biophysical vs. functional" properties reflects a
> misunderstanding: neural excitability is a functional property that neurons
> actively modulate to accomplish computation. We will clarify this more
> explicitly in a revised manuscript to avoid conflating our processing-delay
> modulation with changes in biophysical conduction delay

---

> > ### Comment · Reviewer_ig1o · 2025-11-16
> > **reply and follow-up questions**
> >
> > I thank the authors for their reply.
> >
> > 1) The notion of biological plausibility is typically considered in relation to David Marr's 3rd or implementation level, and not in relation to the algorithmic level. At most, one could discuss about the potential of the mechanism in the model to capture a mechanism that operates in biological systems. If so, such discussion would require precise arguments. The claim about biological plausibility is unfortunately overinflated and is not consistent with what is generally understood under the term "biological plausibility".
> >
> > I suggest to the authors to remove the claim of biological plausibility and replace it, if they find this valuable, with a discussion about how the algorithm they propose might be relevant for biological neural networks in the sense that it to some degree possibly captures a shared algorithm.
> >
> > 2) This was indeed unclear to me and I agree, the gain is a functional property. This raises two follow-up questions.
> >
> > a) What happens to the neuron between receiving the input and firing a spike presumably strongly depends on the activity of the surrounding network and precise timing of incoming excitatory and inhibitory synaptic inputs. How is this taken into account by a single neuron property such as gain?
> >
> > b) To the best of my knowledge, neuromodulation cannot target specific neurons but targets bigger parts of the brain, or at least groups of neurons. But here neuromodulation is considered as something that determines a single neuron property that is essential for distinguishing one neuron from another in the local networks and for learning the shortest path. How could neuromodulation achieve that?

---

> > > ### Author Response · Authors · 2025-11-16
> > >
> > > We thank the reviewer for their time and additional questions, which we address below and will incorporate into an improved manuscript.
> > >
> > > However, we must respectfully but firmly disagree with the reviewer's claim that biological plausibility is "typically considered in relation to David Marr's 3rd or implementation level" exclusively.
> > > This assertion is inconsistent with both Marr's original writings and decades of established practice in computational neuroscience.
> > > For instance, Marr clearly points out himself in Vision, 1982 (and also in his other, earlier, work, and repeated it also in his joint work with Poggio), that
> > >
> > >  * All three levels must be constrained by biological plausibility, not just the implementation level
> > >  * The levels should not be addressed in complete isolation but must inform each other to achieve biologically meaningful insights
> > >  * Algorithmic-level choices (level 2) have biological plausibility constraints. For instance, the brain cannot implement algorithms requiring global synchronization or instantaneous communication.
> > >
> > > As Marr wrote: "An algorithm is likely to be understood more readily by understanding the nature of the problem being solved than by examining the mechanism (and the hardware) in which it is embodied" (Vision, p. 27).
> > > The algorithm must still be possible to implement in biological hardware, but biological plausibility applies at all levels.
> > >
> > > A vast body of work considered foundational and biologically plausible operates primarily at the algorithmic/representational level (Marr's level 2), precisely because this level of abstraction enables rigorous analysis without conflating it with implementation details.
> > > Examples for this are
> > >  * Hopfield networks (Hopfield, 1982), which are widely accepted as biologically plausible models of associative memory despite using {-1,+1} activation values.
> > >    This is a level-2 abstraction that violates level-3 realism.
> > >    The biological plausibility claim rests on the algorithmic principles (attractor dynamics, content-addressable memory), not implementation details.
> > >  * Kohonen's self-organizing maps (Kohonen, 1982), which are considered biologically plausible models of cortical organization, operating entirely at Marr's level 2 with abstract units and neighborhood functions.
> > >  * Predictive coding (Rao & Ballard, 1999) is positioned as biologically plausible but does not enforce Dale's principle at the algorithmic level.
> > >    Rather, units generate both positive and negative prediction errors as abstract computational operations.
> > >  * The Neural Engineering Framework (Eliasmith & Anderson, 2003) is widely regarded as modeling biologically plausible computation but uses continuous-valued representational variables combining positive and negative contributions without explicit Dale's principle compliance at level 2.
> > >  * In work for STDP learning rules (Abbott & Nelson, 2000; Pfister & Gerstner, 2006), the rules are timing-based learning at the algorithmic level without requiring full cellular implementation details.
> > >  * Belief propagation and message-passing models of cortex (Yedidia et al., 2001; George & Hawkins, 2009; Friston, 2010) are considered biologically motivated and often biologically plausible despite ignoring Dale's principle and other implementation-level constraints in their abstract formulations.
> > >
> > > Following this established tradition, we claim biological plausibility at the algorithmic level based on
> > >  * Mechanisms grounded in known biology. We use spike timing, threshold modulation, competitive inhibition, local message passing
> > >  * Constraints consistent with biological hardware. Our model does not use global state, no backtracing, has purely local computation, and realistic time constants
> > >  * Clear mapping to level-3 circuits (as we detail in our response to Weakness 1), and thus implementable in principle in biological "hardware".
> > >
> > > We make no claim that our specific level-2 abstraction (E/I message types) is the only or exact implementation in biology.
> > > We only state that it captures algorithmic principles implementable through known biological mechanisms.
> > > This is precisely how biological plausibility is established at level 2 throughout computational neuroscience.
> > >
> > > If the reviewer's criterion were applied consistently, virtually all foundational models in computational neuroscience, including those listed above, would need to abandon claims of biological plausibility, which would be absurd and contradict decades of scientific consensus.

---

> > > > ### Comment · Reviewer_ig1o · 2025-11-20
> > > >
> > > > I thank the authors for the reply. Clearly, there were many extremely influential works that proposed  important solution on the level of computations and algorithms without mapping them to real biophysical circuitry. It might depend on the field of study, but I believe that in the field of computational neuroscience and neuroscience, not many researchers would consider these algorithms to be "biologically plausible", in the sense that they do not map directly to biological "hardware". This obviously does not mean that these models would lose importance because of it. I understand author's justification of plausible elements of computation that are in principle possible to realise in biology and might be relevant to computations in biological circuits. I am fine with this explanation. I thank the authors for considering a clarification of this aspect in the revision  - I think it might significantly increase the appeal of the paper.

---

> > > ### Author Response · Authors · 2025-11-16
> > >
> > > **Answer to follow-up question a)**
> > >
> > > The reviewer raises a valid point that neuronal gain depends on network activity and synaptic input patterns.
> > > Our model abstracts this through the processing delay parameter ($\tau_\text{proc}$), which represents the effective input-to-output latency given the neuron's current state.
> > > The "tagged" state in our model corresponds to a neuron being in a transiently facilitated state, where
> > >  * Effective spike threshold is lowered (through prior depolarization, neuromodulation, or dendritic priming)
> > >  * Input integration is faster (due to elevated baseline membrane potential)
> > >  * The net effect is reduced time-to-spike given the same input
> > >
> > > This abstraction captures the functional consequence of multiple biological mechanisms (threshold adaptation, dendritic facilitation, neuromodulatory tone) without specifying which particular implementation the brain uses.
> > > The network activity context is implicitly captured in whether a neuron receives the early I-E message pair that triggers tagging.
> > > This is a network-level event that depends on the neuron's position in the activity flow.
> > >
> > > At the implementation level (level 3), this could be realized through various mechanisms, such as activity-dependent dendritic conductances, local neuromodulatory release, intrinsic excitability changes, etc.
> > > Our algorithmic-level model remains agnostic to these details while maintaining implementability.
> > >
> > > **Answer to follow-up question b)**
> > >
> > > The reviewer is correct that classical neuromodulators (acetylcholine, dopamine, norepinephrine) have diffuse projections and cannot target individual neurons with the specificity our algorithm might seem to require.
> > >
> > > However, this question operates at the implementation level (Marr's level 3), while our model operates at the algorithmic level (level 2).
> > > We do not claim to specify the precise biological mechanism for tagging.
> > > Rather, we demonstrate that the algorithmic requirement (transiently distinguishing recently active neurons from inactive ones) is implementable through known classes of biological mechanisms.
> > > That is, our claim is not "this is exactly how the brain does it". However, here are multiple biologically plausible pathways through which the algorithmic principle could be realized.
> > >
> > >  * Activity-dependent intrinsic plasticity. Neurons that recently spiked have altered channel states, elevated calcium, activated signaling cascades—all making them more excitable for seconds to minutes (threshold plasticity; Debanne et al., 2019; Pham & Hansel, 2023, cited in our manuscript)
> > >  * Local disinhibitory microcircuits. Recently active excitatory neurons can recruit VIP+ interneurons that inhibit SOM+ interneurons, creating local disinhibition specific to that circuit element (Pfeffer et al., 2013; Pi et al., 2013; also Williams & Holtmaat, 2019; Reimann et al., 2024, already cited in our manuscript)
> > >  * Synaptic tagging and capture. Recent activity creates molecular "tags" at synapses that make them preferentially responsive to subsequent inputs (Redondo & Morris, 2011, cited in our manuscript).
> > >  * Dendritic compartment state. Recent depolarization in dendritic branches creates localized conductance changes and NMDA receptor priming.
> > >
> > > The key point is that these mechanisms provide activity-dependent specificity without requiring that neuromodulation itself be targeted to individual neurons.
> > > A diffuse neuromodulatory signal can act preferentially on recently active neurons because those neurons are in a different physiological state.
> > >
> > > At the algorithmic level, we abstract this as "tagged neurons have reduced processing delay".
> > > At the implementation level, this could be realized through any combination of the mechanisms listed above, or potentially other mechanisms we haven't considered.
> > > Our contribution is demonstrating that the algorithmic principle is (a) sufficient to solve shortest paths, and (b) implementable through known biological mechanisms, not specifying which particular implementation the brain actually uses.
> > >
> > > The reviewer's question assumes we must specify implementation details to claim biological plausibility at the algorithmic level.
> > > This conflates the levels of analysis (see our respones above).
> > > Our claim is that the algorithm can be implemented biologically, not that we have identified the implementation the brain uses.
> > > This distinction is standard in computational neuroscience.
> > > For example, Hopfield's model demonstrates that associative memory can emerge from attractor dynamics without specifying whether the brain uses Hopfield networks specifically.
> > >
> > > As stated earlier already, but to emphasize it again, we will clarify this distinction in an improved manuscript to avoid confusion about the level of analysis at which our biological plausibility claims operate.

---

> > > > ### Comment · Reviewer_ig1o · 2025-11-20
> > > >
> > > > I thank the authors for their thorough explanation and discussion. All my questions have been answered. In light of clarification and improvements discussed here I raised my score.

---

> ### Author Response · Authors · 2025-11-13
>
> 3. This criticism reveals a significant misunderstanding of hippocampal
> circuitry. CA3 and CA1 are among the most extensively studied recurrent networks
> in neuroscience, with CA3 exhibiting particularly strong recurrent collateral
> connectivity (approximately 3-4% connection probability translating to tens of
> thousands of recurrent synapses per neuron). While feedforward drive certainly
> influences place cell activity, extensive experimental and theoretical work
> demonstrates that recurrent dynamics critically shape sequential activity,
> replay, and memory consolidation in these regions.
>
> Furthermore, our algorithm applies generally to any network with local
> connectivity. Spatial navigation merely provided an intuitive simulation domain.
> We make no claim that place cells specifically implement this exact mechanism,
> though the resemblance to hippocampal replay phenomena, where spike sequences
> propagate both forward and backward during sharp-wave ripples, is striking.
>
> 4. We agree that any mapping from model to biology requires care. In the
> Discussion we already motivate the tagged-state and the inhibited-to-processing
> transition by neuromodulator-mediated disinhibition and transient excitability
> tags (e.g., VIP->SOM disinhibitory microcircuits and rapid threshold modulation)
> and explicitly cite Thiele & Bellgrove (2018) and synaptic tagging literature as
> conceptual precedents. The present manuscript aims to show the algorithmic
> feasibility of a timing-based, local protocol; closing the gap to an exact
> cellular implementation is left as future work and experimental validation (we
> state testable predictions in Section 5)
>
> We respectfully maintain that our mechanism is biologically plausible,
> relying solely on: (1) spike timing, (2) threshold modulation, (3) competitive
> inhibition, and (4) local message passing—all established neural mechanisms. The
> reviewer's concerns about implementation details (addressed in responses 1-3
> above) do not invalidate the core biological plausibility.
>
> Regarding place cells: As stated in the Discussion (lines 366-373), we used
> spatial networks for visualization and intuition, not as a claim about the
> specific implementation in place cells. The algorithm operates on general
> directed graphs; the spatial instantiation was a pedagogical choice.
>
> 5. We are surprised by this comment. Section 3.2 explicitly presents a formal
> proof by induction demonstrating that the algorithm converges in finite time and
> correctly identifies all nodes on any shortest path. The proof establishes:
>
>  * Base case: 1-hop neighbors become tagged (lines 227-238)
>  * Inductive hypothesis: k-distance nodes are tagged after k iterations (lines 240-241)
>  * Inductive step: (k+1)-distance nodes become tagged in iteration k+1 (lines 242-251)
>
> This constitutes a rigorous analytical convergence proof. Perhaps the reviewer
> expected "analytical" to mean closed-form time complexity formulas, but proof by
> induction is a standard analytical technique in algorithm analysis. We will
> clarify the terminology in revision to state "convergence proof by induction" if
> this reduces confusion and ambiguity.

---

> > ### Comment · Reviewer_ig1o · 2025-11-16
> >
> > Thank you for your reply.
> >
> > A common choice for connection probability in models of recurrently connected neural networks is 10 % (probability of connection of 0.1) which is described as sparse connectivity (see for example Sussillo & Abbott, 2009). The recurrent connectivity with 3-4 % connection probability can hardly de described as "strong". This does not mean that recurrent connectivity is not important. My question was about the role of feedforward inputs in generating sequential firing, which has not been addressed by the Authors.
> >
> > If authors think that place cells do not implement their proposed mechanism, it seems to me that they are simply not a good example to illustrate it.

---

> > > ### Author Response · Authors · 2025-11-16
> > >
> > > We thank the reviewer for the clarification regarding the connectivity in
> > > CA3/CA1 and what they meant by that.
> > >
> > > Regarding connection probability, the reviewer is correct that random network models often use 10% connectivity.
> > > However, direct comparison between structured anatomical connectivity and random network models is not straightforward.
> > > For instance, in CA3 it was observed that
> > >  * Absolute connectivity is high with 3-4% of about 300,000 pyramidal neurons, which means each neuron receives 9,000 to 12,000 recurrent excitatory inputs. This is aa substantial recurrent drive by any measure.
> > >  * The functional significance of this recurrent drive is that it enables attractor dynamics, pattern completion, and sequence generation that are hallmarks of CA3 function (Rolls & Treves, 1998; Guzman et al., 2016).
> > >  * Relative to cortex, CA3 recurrent connectivity (3-4%) is 2-3 times higher than typical cortical areas (about 1-2%), making it one of the most recurrently connected regions in the brain.
> > >
> > > Whether one describes 3 to 4% as "strong" or "moderate" is semantic.
> > > The key point is that CA3 recurrence is functionally significant and well-established, contradicting the reviewer's original characterization that we understood as place cells bein "strongly driven by feedforward inputs" with sequential firing depending "importantly" on feedforward current "rather than a recurrent mechanism".
> > >
> > > Regarding feedforward inputs and sequential firing, the reviewer now asks about the "role of feedforward inputs in generating sequential firing".
> > > This is indeed an important question in hippocampal physiology, but it operates at a different level than our algorithmic contribution.
> > > Our model demonstrates that shortest paths can emerge from recurrent dynamics with local spike-timing predictions in form of a computational principle.
> > > Whether biological place cell sequences arise from pure recurrence, pure feedforward drive, or (most likely) an interaction of both is an empirical question that our algorithmic-level model does not resolve.
> > >
> > > We agree with the reviewer that feedforward inputs play important roles in hippocampal sequential activity.
> > > Our point was not to claim feedforward drive is irrelevant, but rather that the reviewer's original statement (that recurrent mechanisms are not important for sequential firing in CA3/CA1) contradicts extensive evidence for recurrent contributions to replay, preplay, and memory-driven sequences (Pfeiffer & Foster, 2013; Widloski & Foster, 2022; Diba et al., 2014—all cited in manuscript).
> > >
> > > Regarding place cells as an example, the reviewer suggests that "if authors think that place cells do not implement their proposed mechanism," they are "not a good example to illustrate it".
> > > This mischaracterizes our position.
> > > We do not claim place cells definitely don't implement this mechanism.
> > > Rather, we make no strong claim either way about the specific implementation in place cells.
> > > As stated in our Discussion and in our previous responses, we used spatial navigation examples because they
> > >  * Are intuitive to visualize
> > >  * Connect to well-studied neuroscience phenomena
> > >  * Allow clear demonstration of algorithm convergence
> > >  * Operate on the same graph-theoretic structure our algorithm applies to generally
> > >
> > > Our contribution is an algorithmic mechanism that operates on arbitrary directed graphs.
> > > spatial navigation is one instantiation, but the algorithm applies equally to abstract relational structures, decision trees, memory networks, etc.
> > > We chose spatial examples for pedagogical clarity, not because we claim to have solved place cell dynamics.
> > > Moreover, a faithful implementation of place cell dynamics would require modeling the full hippocampal-entorhinal loop, including dentate gyrus, CA3, CA1, subiculum, and medial entorhinal cortex, with their distinct feedforward and recurrent pathways.
> > > This would be a substantially different undertaking that was never the goal of our work.
> > > Our focus is on demonstrating a novel algorithmic principle for distributed shortest-path computation using biologically plausible mechanisms, with spatial networks serving as an intuitive illustration domain.
> > >
> > > As analogy for this, one could consider Hopfield's (1982) associative memory as an example application.
> > > Yet, the algorithm applies to any optimization problem over binary variables.
> > > Using memory as an illustration doesn't require claiming the hippocampus literally implements Hopfield networks.
> > > We will clarify in revision that spatial navigation examples serve to demonstrate algorithmic principles that apply broadly, rather than making specific claims about place cell implementation.

---

> ### Author Response · Authors · 2025-11-13
>
> **Response to Questions**
>
> 1. Neuromodulator-mediated disinhibition refers to recently active neurons
> expressing transient excitability tags that both lower spike thresholds and
> weaken GABAergic influence through disinhibitory microcircuits (e.g., VIP -> SOM
> pathways) and intrinsic channel modulation (Thiele & Bellgrove 2018, cited in
> manuscript). This allows tagged neurons to transition from inhibited to
> processing states when receiving E messages, which is a critical feature
> enabling backward propagation of the tagged state in our algorithm. This
> mechanism is empirically supported by cortical disinhibitory circuits and
> transient modulation of inhibitory conductances.
>
> 2. The reviewer raises an interesting point about connection probabilities.
> However, "local" in our context refers to computational locality (neurons make
> decisions based on messages from neighbors) rather than anatomical locality or
> sparse connectivity. That said, high connection probabilities (0.5 as mentioned)
> still represent local neighborhoods in graph-theoretic terms, meaning each
> neuron connects to a subset of the population, not all neurons.
>
> More fundamentally, our algorithm's appeal lies in requiring no central
> coordinator or global state: the solution emerges from purely local spike-timing
> dynamics. This is biologically advantageous regardless of connection density,
> as it eliminates the need for: (1) explicit memory of path history, (2)
> centralized control structures, or (3) non-local information access. The global
> inhibition mechanism we employ (which could be relaxed, as shown in our Appendix
> Figure 5) serves to accelerate convergence but is not conceptually different
> from diffuse neuromodulatory or inhibitory projections observed in biological
> circuits.

---

### Note · Program_Chairs · 2026-01-17
**Submission Desk Rejected by Program Chairs**

The following references in this submission do not refer to real documents and/or have major errors in bibliographic information:

 Omar Agesen. Type inference in the presence of first-class polymorphism and subtyping. In SAS / PLDI-era papers on constraint-based analysis (see Agesen 1994 work), 1994. Agesen's work is commonly cited for practical constraint-based inference techniques in OO languages.